# How Chanting Relates to Cognitive Function, Altered States and Quality of Life

**DOI:** 10.3390/brainsci12111456

**Published:** 2022-10-27

**Authors:** Gemma Perry, Vince Polito, Narayan Sankaran, William Forde Thompson

**Affiliations:** 1School of Psychological Sciences, Macquarie University, Sydney, NSW 2109, Australia; 2Faculty of Society and Design, Bond University, Gold Coast, QLD 4229, Australia; 3School of Medicine, University of California, San Francisco, CA 415, USA

**Keywords:** chanting, devotion, mindfulness, altered states, quality of life, mind wandering

## Abstract

Chanting is practiced in many religious and secular traditions and involves rhythmic vocalization or mental repetition of a sound or phrase. This study examined how chanting relates to cognitive function, altered states, and quality of life across a wide range of traditions. A global survey was used to assess experiences during chanting including flow states, mystical experiences, mindfulness, and mind wandering. Further, attributes of chanting were assessed to determine their association with altered states and cognitive benefits, and whether psychological correlates of chanting are associated with quality of life. Responses were analyzed from 456 English speaking participants who regularly chant across 32 countries and various chanting traditions. Results revealed that different aspects of chanting were associated with distinctive experiential outcomes. Stronger intentionality (devotion, intention, sound) and higher chanting engagement (experience, practice duration, regularity) were associated with altered states and cognitive benefits. Participants whose main practice was call and response chanting reported higher scores of mystical experiences. Participants whose main practice was repetitive prayer reported lower mind wandering. Lastly, intentionality and engagement were associated with quality of life indirectly through altered states and cognitive benefits. This research sheds new light on the phenomenology and psychological consequences of chanting across a range of practices and traditions.

## 1. Introduction

Chanting is among the most pervasive meditative practice worldwide, playing a central role in many religious and secular traditions across cultures. It typically involves the repetition of a chosen phrase, word, or syllables, while disregarding distractions [1,2,3]. In some traditions, the sound recited is referred to as a mantra or prayer and is considered a devotional form of music or meditation [4,5]. Chanting can be practiced silently, vocally, alone or in unison with others as a form of synchronous vocalisation and movement. Group chanting can also be practiced using a call and response style whereby one person leads the chant, and a group responds with the same phrase [6].

Chanting has been used throughout the world for thousands of years for worship, ritual, strengthening community and healing [2,5,7]. Chanting is both a ubiquitous and ancient practice with many traditions such as Buddhism, Sufism, Hinduism, and Yogic traditions believing chanting to be a way of altering states of awareness and reaching full human potential [2,8,9,10]. Indigenous Australians use chanting to communicate with ancestors and spiritual beings [11]. In the United States, the Navaho chant to prevent illness and in sacred celebration ceremonies [12]. In India, the chanting of ancient texts is a form of spiritual worship and may be used to transcend awareness [4,8]. Jewish cantillation is another form of chanting and is used for transformation and worship [13]. Despite the historical and global prevalence of chanting, its psychological effects are poorly understood.

Chanting has been found to decrease stress and depressive symptoms [14,15,16], increase focused attention [17], increase social cohesion [18], and induce mystical experiences [19]. However, the full impact of chanting has been underexplored and there may be additional psychological effects. Further, it is unclear how chanting impacts overall quality of life. The current investigation surveyed the phenomenology of chanting across 32 countries and various chanting traditions, focusing on how intentionality and engagement during chanting impact upon a range of outcomes and experiences, including flow, mystical experiences, mindfulness, mind wandering, and quality of life.

The phenomenology of chanting is a largely unexplored area of research. Further, little is known about how specific attributes of chanting interact with psychological benefits. Although chanting is both a spiritual and secular practice, previous research has focused on secular forms of chanting. For example, Transcendental Meditation (TM), a term coined to distinguish the practice from Hinduism and separate it from any religion [20], is likely the most widely studied chanting practice, and involves the silent repetition of a sound [21]. TM is sometimes referred to as a religion [22], but the movement has focused on secularising chanting, arguably to gain approval from biomedical communities and attract Westerners [20,23,24]. It is unclear whether the benefits of chanting depend on individuals holding any specific beliefs, but many contemplative practices come from rich religious and spiritual traditions, so it is possible that such beliefs are relevant to the impact of chanting. For example, TM emerged from the Vedic tradition of India [8], and many mindfulness practices are inspired by Buddhism [24]. Until now, these practices have been researched and promoted as nonreligious and nonesoteric [25]. By contrast, in the current study, chanting is acknowledged as a holistic contemplative practice that comes from rich traditions, many of which involve devotional qualities. Specifically, properties of chanting practice such as a focus on sound, devotion, or intention while chanting may yield or interact with experiential outcomes of chanting such as altered states, cognitive benefits, and quality of life.

Further, research has not explored whether different types of chanting practice give rise to distinct psychological outcomes. For example, chanting can be done as a form of individual, silent, repetitive prayer or as a vocal group practice including dancing and clapping. It is unclear whether varied practices provide unique benefits. Different forms of chanting involve specific mechanisms, such as intense concentration for a silent individual practice, or complex rhythmic synchrony for a vocal group practice. It is likely the emphasis on diverse mechanisms could give rise to distinct psychological outcomes.

## 2. Intentionality

Chanting is sometimes associated with religious beliefs, but is also practiced in the absence of beliefs. This observation suggests that the benefits of chanting may not rely solely on the presence of specific religious beliefs, but rather on a broader concept of focused intentionality: an emphasis during chanting on a particular point of focus. Indeed, all chanting traditions tend to focus on one or more of sound, devotion, and intention. To clarify, intentionality is used here in the philosophical sense to mean that consciousness is directed towards something. An intention refers to any contemplation of a goal, aspiration, or affirmation. These three types of focus (sound, devotion and intention) vary in emphasis depending on the practice and tradition. Some traditions place an emphasis on the sound itself, and the precise pronunciation of words (as in some forms of Vedic Chanting) [9]. In other traditions, an emphasis is placed on the concept of devotion (as in Bhakti Yoga traditions) [26]. In some practices, it is also common to form an intention statement (i.e., a goal, aspiration, or affirmation) of the practice (referred to as a ‘sankalpa’ in the yogic tradition). Intentions often take the form of a short, positive statement [27], such as “I am healthy” or “I am confident.” Intentions during chanting may also be contemplated in the form of a prayer. For example, in some Buddhist practices, chanting is done to promote compassion or peace. The experiential impact of intentionality during chanting is largely unknown. The current investigation attempts to determine the extent to which these various forms of focus in chanting impact upon experiential outcomes.

### 2.1. Sound

Focusing on the sound of chanting could have psychological benefits that are independent of beliefs or intentions. Specific patterns of neurological activity have been observed during the act of mentally or vocally repeating a sound, as in the practice of chanting. For example, clusters of activation are found in areas of the brain involved in speech production and attention regulation, such as the motor control network [28]. Additionally, significant cluster deactivations can occur in areas related to awareness of body sensations, such as the anterior insula [28]. Therefore, the simple act of repeating a sound may increase focused attention while decreasing awareness of bodily sensations. Further, if the repetition of the sound is vocalized, slowed breathing is likely to activate the parasympathetic nervous system [29]. These effects suggest that even if one is distracted, vocalisation is likely to have positive respiratory and hormonal changes that may contribute to feelings of relaxation and positive mood [30]. Therefore, feelings of relaxation may arise from vocalising, and focusing on the sound and vibration created while chanting thereby decreasing stress through focused attention and physiological changes related to the parasympathetic nervous system. 

### 2.2. Devotion

Some traditions emphasize the importance of devotion, whereby concentration on the meaning associated with the sound must be maintained throughout the practice [9]. There are sounds related to spiritual symbols such as deities, objects, or qualities such as wisdom, strength, abundance, perseverance, or compassion [2,5]. For example, certain mantras in the Hindu tradition, such as “Om Gam Ganeshaya Namaha”, are thought to call upon the energy of the Hindu deity Ganesh, thereby helping to overcome obstacles and organize information in the mind [31]. Another example is the mantra “Om Mane Padme Hum” from the Buddhist tradition associated with Avalokiteshvara, a form of Buddha signifying compassion [5]. These types of appraisals while chanting can provide people with a sense of meaning and belonging as they are reminded that they are part of something larger than themselves (e.g., a belief system, group, deity) [32]. This may also facilitate feelings of surrender, connection and assist with redirecting attention away from trivial everyday life stressors and toward spiritual beliefs. 

It is possible that the psychological benefits of chanting are amplified by focusing attention on the meaning of what is being chanted. Research suggests that spiritually meaningful phrases may be especially important [16,33,34]. Wachholtz & Pargament [33] found that using spiritually meaningful phrases during meditation had a greater impact on psychological, spiritual and pain tolerance variables when compared to secular phrases. Presumably, the feeling states associated with meaningful sounds may enhance the benefits of chanting. For example, feelings of surrender during prayer are associated with decreased activation in the frontal lobe, an area related to focused attention [35]. Usually, attentional practices such as centering prayer and Buddhist chanting are associated with increased activity in areas of the brain related to attention (such as the prefrontal cortex). However, when measuring brain regions in participants who included the act of surrender to God while chanting, decreased activation in these attention related areas has been observed. Similar brain activity has been reported for individuals who are speaking in tongues, and mediums when they are ‘channeling’ information [36].

Newberg et al. [35] measured feelings of surrender using single photon emission computed tomography (SPECT) to assess whether there were cerebral blood flow (CBF) changes during Islamic prayer. Two participants recited the Dhikr prayer and were measured at rest and while praying. One participant recited the Salat prayer (Muslim prayer done 5 times per day) and was measured while doing the prayer ‘automatically’ and then while doing the same prayer with intention, which included the intention of surrendering to God. Decreased frontal lobe activity was found in all participants and all prayers. Islamic prayer practices showed a decrease in CBF in the prefrontal cortex and related frontal lobe structures, as well as in parietal lobes. However, the Salat intentional prayer increased CBF in other areas that the Dhikr prayer did not (specifically in the anterior cingulate gyrus, caudate nucleus insula, thalamus, and globus pallidus). Similar increases in CBF have been found in a group of Danish Christians during silent religious prayer and resulted in positive emotional experiences [37]. Although this is still preliminary research, overall, these findings indicate that devotion may enhance benefits of chanting beyond the effects of vocalization, or focus on sound (e.g., pronunciation or vocalization), on the parasympathetic nervous system. 

### 2.3. Intention

Chanting can also be practiced with an intention in mind, either in a devotional or non-devotional setting. Intentions can be goals, aspirations, or affirmations. For example, a practitioner might set a goal at the beginning of a chanting ritual and chant with that goal in mind. Goals can be individual or communal and may augment the psychosocial impact of chanting. In the case of individual goals, having an intention for the practice may enhance attention, making the practice more meaningful. The more one can associate with an action, the more salient the experience [32]. Communal goals may also intensify the chanting experience as shared attention can generate perceived emotional synchrony in the individual which in turn can foster feelings of social connection and oneness with group members [32]. Further, intentions may be reinforced through shared goals and attention might be enhanced lending to full embodied participation in chanting practices. Taken together, individual or group intention may lead to enhanced effects of chanting. 

## 3. Chanting and Altered States: Mystical Experiences and Flow States

One of the potential outcomes of chanting is the experience of altered states of consciousness. Research suggests chanting is associated with alterations in consciousness such as mystical states [19]. However, other potential states have not yet been explored, and it is unknown how such states interact with the cognitive impact of chanting or perceived quality of life. It is also unknown whether such altered states of consciousness are more likely to occur for some traditions of chanting over others. 

Mystical states are typically reported as extreme positive alterations of consciousness, and often occur in contemplative practices [38], group rituals [39], and during engagement with music and dance [40]. However, there is little research on their association with chanting. Chanting should give rise to mystical states as it can be considered a contemplative, musical and synchronous ritual. Persisting positive changes in sense of purpose, perceived quality of life, and life satisfaction have been found to be related to mystical experiences [41]. In a survey on ‘God encounter’ events, mystical experiences were rated as among the most meaningful and significant events in one’s life, leading to persisting positive changes in overall life satisfaction, meaning and purpose [42]. In view of such findings, it is likely that mystical experiences arising from chanting may also give rise to enhanced perceived quality of life. 

Similarly, flow states are often evoked by music listening, movement or synchronous activity [43] and might also be associated with chanting. Flow is a state in which an individual becomes so involved in an activity that everything else seems to become irrelevant [44]. There are a few reasons that chanting might be associated with states of flow. Firstly, the collective ritual of chanting can increase endorphin production which may be related to feelings of flow [32,45]. Secondly, flow states have been found to occur during periods of intense focus [46]. Chanting may be a form of focused attention depending on how it is used, and many traditions emphasize the need for deep concentration on the sound or meaning of chanting. Lastly, flow states have been found to occur during challenging tasks [47]. Therefore, if chanting involves complex, simultaneous rhythms, (as in the case of traditions like Taketina that include stepping, clapping, and chanting in three different rhythms), then flow states could occur from the chanting being a challenging task. Taken together, chanting may induce flow states through biochemical changes, focused attention, and challenging rhythmic patterns.

## 4. Cognitive Benefits of Chanting: Mindfulness and Mind Wandering

Chanting may also be associated with certain cognitive benefits such as increased mindfulness and decreased mind wandering. Mindfulness is an explicit component of many traditions, both as a long-term goal of chanting (to become more objectively aware in the present moment day-to-day) and as a technique used during chanting (practicing detachment and objective awareness of feelings and thoughts while chanting). 

Mindfulness is the objective awareness in the present moment and was popularized in the West by Jon Kabat-Zinn, originally inspired by the Buddhist tradition [33]. However, it is important to note, the concept of mindfulness exists within many traditions and practices [48]. For example, in yoga practices, objectively observing thoughts, emotions, breath or a mantra is referred to as the witnessing state, witness consciousness or the seer [10,49]. Although mindfulness is a goal of many chanting traditions, there is little research on the relationship between chanting and mindfulness. Bormann [17] found mindful attention to increase with a mantram repetition program over a 5-week period. However, it is not known how mindfulness is related to chanting over longer periods of time, across varied traditions, and how mindfulness interacts with other experiential outcomes of chanting such as altered states and quality of life.

Mind wandering, on the other hand, is the inability to attend to the present moment and is commonly associated with states of depression [50,51]. Chanting has been found to increase focused attention which should decrease mind wandering [17,18]. However, to our knowledge, there is no research that directly investigates chanting and mind wandering. It is important to evaluate whether chanting is associated with enhanced mindfulness and decreased mind wandering, which in turn may relate to one’s overall mental health and quality of life.

Aspects of chanting may be associated with mindfulness and mind wandering. For example, if chanting is practiced silently as in the form of mentally repeating a sound, phrase, or prayer, this could give rise to higher mindfulness and lower mind wandering compared to if chanting is practiced vocally in a group conversational style. Mental repetition of a sound without vocalising might require higher levels of concentration and discipline which in turn may promote more mindfulness or reduced mind wandering [52]. In summary, there may be certain aspects of chanting that may enhance effects of mindfulness and decrease mind wandering. 

## 5. Chanting and Quality of Life

Quality of life encompasses psychological, physiological, and social aspects. Chanting could impact quality of life through social implications of group chanting, physiological benefits from chanting practices such as enhanced cognitive skills, and reduced stress from the effect on the parasympathetic nervous system. Moreover, chanting may have indirect effects on quality of life by catalysing experiences of altered states (i.e., mystical experiences and flow) and cognitive benefits (i.e., reduced mind wandering and increased mindfulness). 

Altered states are often reported as being some of the most meaningful experiences of people’s lives [41]. Altered states often include feelings of connection to others and/or nature [53,54,55]. This could improve personal relationships and how one considers the environment. In this way, if chanting is associated with alterations in consciousness, that in turn, could enhance overall perceived quality of life.

Mindfulness has also been found to be associated with improved quality of life, decreased anxiety and depressive systems while mind wandering is associated with depression and anxiety [51]. Mindfulness emphasizes that the individual is more than just thoughts which can assist in detaching from intrusive thoughts and feelings. Therefore, if chanting is associated with cognitive benefits such as increased mindfulness and decreased mind wandering, that in turn could enhance perceived quality of life.

## 6. Aim of Survey

Evidence suggests that chanting can confer several psychological benefits, but questions remain about the phenomenology, nature and causes of such benefits. Is intentionality toward sound, devotion, and intention important in determining psychological benefits of chanting? Is higher engagement with chanting associated with altered states or cognitive benefits? Do different styles of chanting contribute to diverse benefits? How do intentionality, chanting engagement, altered states and cognitive benefits interact with each other and impact overall quality of life?

To unpack these issues, the current study involved an online survey, which investigated chanting practices from various traditions. Questionnaires explored phenomenology of chanting, types of chanting practices, and perceived benefits.

The study had 3 main aims. The first aim was to examine the relationship between attributes of chanting practice (intentionality, and level of engagement) and psychological outcomes (altered states of consciousness and cognitive benefits). The second aim was to determine if attributes of chanting practice (intentionality and engagement) and psychological outcomes (altered states and cognitive benefits) were associated with quality of life. Lastly, given the various emphasis on specific mechanisms in different chanting practices, the third aim was to explore if varied chanting styles give rise to specific psychological effects. Considering the current understanding of music and contemplative practices, predictions on the impact of chanting were established. First, greater intentionality (i.e., greater focus on sound, devotion, and intention) and engagement (greater experience, longer practice duration and regularity) should lead to higher scores on measures of altered states (i.e., mystical experience and flow) and cognitive benefits (i.e., increased mindfulness and lower mind wandering). Second, intentionality, engagement, altered states and cognitive benefits should collectively be related to higher quality of life. Finally, we predicted that differential outcomes may occur depending on the structural characteristics of chanting practices, so we included an exploratory comparison of the effects of different styles of chanting on psychological outcomes. 

## 7. Materials and Methods

### 7.1. Recruitment

Participants between the age of 18 and 80 years who engaged in a regular chanting practice and were fluent in English were recruited through social media, community newsletters and notice boards. The survey was posted in online chanting forums as well as distributed through chanting communities. All participants were offered the chance to enter a prize draw to win one of five AUD$100 Amazon vouchers. Ethics approval was given by the Macquarie University Ethics committee and each participant provided consent electronically.

### 7.2. Materials

Participants provided consent before answering demographic questions and reporting on their style of chanting practice. They then completed a series of psychometric scales presented in counterbalanced order. This study was part of a larger project that also included measures of absorption, altruism, religiosity, and traditions experiencing mystical states [19].

### 7.3. Measures

#### 7.3.1. Mindfulness

Mindfulness, the tendency to be aware and attentive of experiences in daily life, was measured using The Mindful Attention Awareness Scale (MAAS) [56] which consists of 15-items asking participants to indicate how frequently they experience behaviors and rate their level of agreement of statements. Each statement is rated on a 6-point Likert-type scale ranging from 1 (almost always) to 6 (almost never). Example statements include “I find it difficult to stay focused on what’s happening in the present” and “I find myself preoccupied with the future or the past”. Scores range from 15 to 90 with higher scores reflecting higher levels of mindful awareness. The MAAS has shown good reliability and validity [57]. Internal consistency for the current study was excellent (α = 0.91).

#### 7.3.2. Mind Wandering

Mind wandering was measured using The Mind Wandering Questionnaire (MWQ) [58], a 5-item measure asking participants to rate their level of agreement on statements regarding attention. Statements are rated on a 6-point Likert-type scale ranging from 1 (almost never) to 6 (almost always). Example statements include “I do things without paying full attention” and “while reading, I find I haven’t been thinking about the text and must therefore read it again”. Scores range from 6 to 30 with higher scores reflecting higher levels of mind wandering. The MWQ has shown convergent validity with existing mind wandering measures as well as high internal consistency [58]. Internal consistency for the current study was good (α = 0.85). 

#### 7.3.3. Engagement

Participants were asked how many months they had been chanting for (experience), how long a typical chanting session lasted (practice duration) and how regularly they chant (regularity). Participants were asked to report the duration of a typical chanting practice, with three categorical response options: 10 min or less, between 10 min and 1 h, 1 h or more. Participants were asked to report how regularly they chant, with four categorical response options: less than once a month, one or two times per month, once per week, once or more per day.

#### 7.3.4. Intentionality

Participants were asked to rate how important sound, devotion and intention were in their chanting practice on a 5-point Likert scale ranging from 1 (not at all important) to 5 (extremely important).

#### 7.3.5. Mystical States

Mystical states were measured using The Revised Mystical Experience Questionnaire (MEQ30) [59], a 30-item measure developed from the MEQ43 [60]. The MEQ30 includes 4 subscales: mystical, positive mood, transcendence of time and space, and ineffability. The MEQ30 was adapted for this study by asking “Looking back on your experiences with chanting, please rate the degree to which at any time during chanting practices you experienced the following phenomena.” Statements were rated on a 6-point Likert scale ranging from 0 (none, not at all) to 5 (extreme). Example items included “Experience of amazement” and “Experience of unity with ultimate reality.” Scores range from 0 to 150 with higher scores reflecting higher mystical experiences. The MEQ30 has good reliability for all 4 subscales [59,60]. Internal consistency for the current study was excellent (α = 0.97). Detailed analysis of the predictors and prevalence of mystical experiences, as well as trait characteristics associated with mystical states, were reported in Perry et al. [19]. In the current study, we investigated the relationship between MEQ scores, mindfulness, mind wandering, flow, intentionality, engagement, and quality of life.

#### 7.3.6. Flow

Flow was measured using The Short Flow State Scale (SFSS) [46], a 9-item scale with statements rated on a 5-point Likert scale ranging from 1 (never) to 5 (always). We adapted this scale for the current study by asking, “Think about how often you experience each characteristic during chanting, then indicate on the scale below which best matches your experience.” Items include statements such as “I do things spontaneously and automatically without having to think” and “the way time passes seems to be different from normal.” Scores range from 9 to 45 with higher scores reflecting a more intense experience of flow. The Short Flow Scale has been found to be adaptable in a variety of settings such as music, movement, and work, and has good reliability and validity [46]. Internal consistency for the current study was acceptable (α = 0.79).

#### 7.3.7. Quality of Life

There were 12 Quality of Life questions (QoL), adapted from Penman et al. [26]. Questions asked how people believed chanting had impacted their life in areas such as mental or emotional health, relationships, self-awareness, life purpose and life satisfaction. Participants were asked to “Please indicate how you believe your chanting practice has influenced your quality of life in the following areas.” Qualities were rated on a 7-point Likert scale ranging from 1 (much worse) to 7 (much better). Scores ranged from 1 to 84 with higher scores indicating higher overall quality of life. The 12 questions were related to physical health, mental health, emotional health, spiritual health, loneliness, positive emotions, sleep, self-esteem, self-awareness, life purpose and life satisfaction.

## 8. Results

### 8.1. Descriptive Statistics

#### 8.1.1. Participants

Of the 707 people who clicked on the survey link, 83 discontinued immediately, whereas 168 commenced the survey but did not complete it. Of those who commenced the survey but did not complete it, they answered an average of 38% of the survey questions before discontinuing. Finally, 456 completed the survey and were included in the final analyses. Ages ranged from 18 to 78 years (*M* = 48.3, *SD* = 12.75). Participants were 75.2% female and 24.8% male. Participants came from 32 countries. The most represented countries included Australia (45.8%), the United States (27.2%), the United Kingdom (4.6%), Canada (4.6%), India (3.3%), Germany (2.4%), the Netherlands (1.8%), Austria (1.8%), and New Zealand (1.1%), with the remainder (7.4%) from countries with less than 1% of participants. Nationalities consisted of Australian (38.6%), American (25.2%), British (4.8%), Canadian (4.6%), Indian (3.5%), German (2.9%), (Dutch 2%), New Zealander (1.7%), Brazilian (1.3%), French (1.1%), Malaysian (1.1%) with the remainder of nationalities (13.2%) with less than 1% of participants (See Appendix A for a detailed distribution of countries). 

#### 8.1.2. Chanting Traditions

Participants indicated the chanting tradition they practiced most. The final sample included practitioners of Vedic (22.6%), Hindu (14.2%), Buddhist (10.1%), Yoga (9.7%, consisting of Kundalini Yoga, Satyananda Yoga, and Sivananda Yoga), Hare Krishna (7%), TaKeTiNa (5.9%), Tantra (5.7%), Silent Mantra (5.7%, consisting of Transcendental Meditation, Primordial Sound Meditation and Vedic Meditation), Kirtan (5.3%, consisting of Kirtan and Bhakti), and other forms of chanting (13.8%). Participants nominated one chanting tradition, however, it is important to acknowledge that some of these traditions have overlapping themes and practices.

The included sample consisted of Monotheistic religions (14%, consisting of Christianity, Judaism and Islam), Dharmic religions (11.8%, consisting of Hinduism and Sikhism), Buddhist (16%), Secular (12.9%), Multi Faith (9.4%, which included participants that listed many faiths, for example ‘Yogic, Hinduism, Buddhism’ or that wrote ‘mixed’ or ’multi faith’, Spiritual (14.3%, which included ‘spiritual’ ‘higher power’ ‘new age’ ‘nature’ and ‘love’, Vedic traditions (7.9%, which included ‘ISKCON’, ‘Yoga’, ‘Vaishnavism’, ‘Vedic Teaching’ and ‘Tantra’, and other (13.6%).

#### 8.1.3. Intentionality and Engagement

Participants were asked to rate how important sound, devotion and intention were on a scale from 1–5, where 5 indicated the highest level of importance. Intention was rated as the most important (*M* = 4.37*, SD* = 0.83), followed by devotion (*M* = 3.98, *SD* = 1.19) and sound (*M* = 3.37, *SD* = 1.22). 

Participants reported chanting experience ranging from 1 to 725 months (*M* = 129, *SD* = 134.60). Overall, participants reported high levels of chanting engagement in the last 12 months, with 62.7% reporting they chanted once or more per day, 24.1% once per week, 9.6% one to two times per month, and 3.5% less than once per month. When asked the length of each chanting session, most participants reported chanting between ten minutes and one hour (53.1%), followed by one hour or more (25.9%) and ten minutes or less (21.1%). 

#### 8.1.4. Chanting Style and Contexts

Participants reported the style of chanting they most often engaged in with most reporting repetitive prayer (57%), followed by call and response (33.3%), and other (9.6%). The means and standard deviations of call and response chanting and repetitive prayer for all five of the outcome variables can be found in Table 1. Lastly, most participants reported chanting vocally (75.9%) compared to silently (24.1%).

### 8.2. Relationship between Attributes of Chanting and Psychological Outcomes

Pearson’s correlations were examined to assess the strength of associations between attributes of chanting practice and psychological outcomes. Specifically, we wanted to assess if greater Intentionality (Sound, Devotion, Intention) and Engagement (Experience, longer Practice Duration and more Regularity) were associated with higher scores of Altered States (Mystical Experience and Flow), Cognitive Impact (Mindfulness and Mind Wandering) and Quality of Life. Pearson correlations are displayed in Figure 1.

### 8.3. Quality of Life

Next, a multiple linear regression was used to determine how all variables were related to Quality of Life. Regularity, Mystical Experience, Flow, and Mindfulness scores had the largest modulatory effects on Quality-of-Life scores in the multiple regression. In each case, there was a positive relationship, whereby higher predictor scores yielded higher Quality of Life scores. This model was significant (*F* = 30.7; *p* = 5.5 × 10^−45^), with the 10 variables together explaining over 40% of the variance in Quality of Life (full model R^2^ = 0.41). The regression coefficients of all statistical predictors of Quality of Life are displayed in Figure 2A (See Appendix A for a detailed regression table).

Figure 2A shows regression coefficients of ten predictors of Quality of Life. Significance was evaluated against a null distribution generated via permutation tests, indicated by the grey markers (* *p* < 0.05; ** *p* < 0.01; *** *p* < 0.001). Figure 2B shows the unique variance in Quality of Life explained by the four families of variables: Engagement (Experience, Practice Duration, Regularity), Intentionality (Sound, Devotion, Intention), Altered States (Flow, Mystical States), and Cognitive Impact (Mindfulness, Mind Wandering).

To further characterize how the different types of variables were uniquely associated with outcomes in Quality of Life (i.e., controlling for the correlations amongst variables displayed in Figure 1), we used a variance partitioning approach to quantify the unique variance in Quality of Life scores explained by four families of variables: Intentionality (comprised of Sound, Devotion and Intention); Engagement (comprised of Experience, Practice Duration and Regularity); Altered States (comprised of Mystical Experience and Flow); and Cognitive Impact (comprised of Mindfulness and Mind Wandering). Specifically, we computed the difference in R^2^ values between the “full model” (the regression model with all predictor variables) and the “reduced model” (the same model but with the variables of interest omitted). Figure 2B shows the results of this procedure for the four different predictor types. Consistent with the regression weights, variables related to Altered States offered the largest predictive power in explaining Quality of Life, above and beyond what other variables could explain.

Intentionality and Engagement were not able to explain unique variance in Quality-of-Life scores. One explanation is that their effects on Quality of Life are indirect, acting by way of intermediate variables. Given altered states and cognitive benefits have been found to be associated with quality of life, a path analysis was implemented to examine whether altered states and cognitive impact mediate the effects of Intentionality and Engagement on Quality of Life. This exploratory path analysis was conducted to examine indirect effects of Engagement (Experience, Practice Duration, Regularity) and Intentionality (Sound, Devotion, Intention) through Altered States (Mystical Experience and Flow) and Cognitive Impact (Mindfulness and Mind Wandering) to perceived Quality of Life.

Figure 3 shows coloured arrows indicate significant effects while nonsignificant effects are indicated by grey arrows. Covariance amongst variables within the same level of the model were omitted for the purposes of visualization but included as parameters in the model. Overall, the model was significant, with largest effects being that of Practice Duration and Devotion on Mystical Experience. Thus, despite Engagement variables (such as Practice Duration) and Intentionality variables (such as Devotion) not having a large predictive effect directly on Quality of Life, we observe that they do predict experiential variables such as Mystical Experience or Flow states, which in turn were able to explain Quality of Life scores. 

The model was a relatively good fit for the data ꭓ^2^451(6) = 39.2, *p* = <.001; RMSEA = 0.11, pclose = 0.001; CFI = 0.956; TLI = 0.706; SRMR = 0.033. Flow predicted Quality of Life primarily from Intention (β = 0.243 [0.14, 0.34] z = 4.74, *p* = < 0.001) and Practice Duration (β = 0.119 [0.03, 0.21] z = 2.64, *p* = 0.008). Mystical Experience predicted Quality of Life primarily from Practice Duration (β = 0.161 [0.07, 0.25] z = 3.63, *p* = < 0.001), Sound (= 0.996 [0.12, 0.19] z = 2.23, *p* = 0.026) and Devotion (β = 0.227 [0.13, 0.33] z = 4.38, *p* = < 0.001). Mindfulness predicted Quality of Life primarily from Experience (β = 0.125 [0.04, 0.21] z = 2.77, *p* = 0.006), Regularity (β = −0.135 [−0.23, −0.40] z = −2.78, *p* = 0.006) and Intention (β = 0.191 [0.09, 0.29] z = 3.67, *p* = < 0.001). Mind Wandering predicted Quality of Life primarily from Experience (β = −0.106 [−0.20, −0.01] z = −2.27, *p* = 0.024) and Regularity (β = 0.142 [0.04, 0.24] z = 2.83, *p* = 0.005).

### 8.4. Chanting Style: Call and Response vs. Repetitive Prayer

To assess differences between call and response and repetitive prayer on Altered States, Cognitive Impact and Quality of Life, one *t*-test for each of the variables was conducted. Participants who indicated they practice a chanting style in the ‘other’ category were excluded leaving a total of 412 participants for this analysis (call and response *n* = 152, repetitive prayer *n* = 260). The five *t*-tests performed are reported in Table 2. Participants who engaged in repetitive prayer were lower on Mind Wandering (*M* = 15.1, *SD* = 4.48) than participants who engaged in call and response (*M* =16.2, *SD* = 4.86). Participants who engaged in call and response chanting were higher on Mystical Experience scores (*M* = 108.9, *SD* = 24.1) than those who engaged with repetitive prayer (*M* = 103.6, *SD* = 27.15). No differences were found between call and response and repetitive prayer for Mindfulness, Flow or Quality of Life.

## 9. Discussion

The current study examined experiences of chanting by surveying 456 individuals involved in a wide range of chanting traditions. Results showed higher intentionality (greater focus on sound, devotion, and intention) and engagement (more experience, practice duration and regularity) were associated with altered states and cognitive benefits. Further, intentionality and engagement impacted quality of life through a range of altered states and cognitive benefits. There were also some differences found between styles of chanting. Call and response chanting was associated with higher scores of mystical experiences compared to repetitive prayer, which was associated with comparatively lower scores of mind wandering. Taken together, these results suggest different levels of intentionality and engagement, and distinct styles of chanting may influence psychological outcomes.

### 9.1. Intentionality

Greater intentionality (sound, devotion, intention) was associated with higher scores on altered states and cognitive impact measures. There are three potential mechanisms that may explain these findings: increases in focused attention; enhanced emotional engagement; and proficiency in chanting practices. First, chanting has been classified as a form of focused attention meditation as it includes focus on the sound which is repeated vocally or silently [14,61]. Bormann et al. [14] has suggested the primary mechanism of chanting is directing attention toward the chosen sound and away from intrusive thoughts. Benson, [62,63] confirms this by stating the two basic steps to repetitive prayer (a form of silent chanting) are (1) repetition of the sound (2) disregarding other thoughts. Focused attention meditation has been found to reduce anxiety and emotional distress as well as improve concentration [64]. If importance is given to intentionality during chanting, this would likely enhance the attentional effects of chanting. 

Second, greater intentionality might enhance emotional engagement and increase the salience of chanting practices. Indeed, if someone is focused on devotion and intention, the practice will be more meaningful than simply reciting sounds. Research has found when performed behaviors are valued, they are attention demanding, memorable, and the experience is more salient [32]. Having an intention while chanting may also create an expectancy effect thereby enhancing engagement. Individuals are more likely to engage with an activity if they expect it will bring behavior or health changes [65]. Further, response expectancy theory states that experience depends partially on an individual’s expectations [66]. Therefore, if an individual is chanting with an intention (expectation) in mind, then they may experience enhanced effects due to expectation compared with an individual repeating sounds without expectations in mind. Lastly, if there is concordance between an activity and the motivation for doing the activity, individuals tend to experience enhanced effects [67]. 

Third, intentionality may enhance effects of chanting due to the cumulative effects of learned chanting proficiency. For example, one can chant by simply repeating sounds, however, one can repeat sounds while focusing on details of the sound (importance of sound such as vibrations, tone, pitch, pronunciation), how the sound is personally meaningful to them (devotion) and what they intend for the sound to do (intention). This could create a cumulative effect of different active mechanisms which could lead to enhanced effects of chanting. Indeed, a review that evaluated 18 comparative studies and 16 meta-analyses on diverse components of yoga found that combined practices outperformed simple interventions [68]. Further, Matko & Sedlmeier [69] found that including ethical aspects of yoga was more effective than simply mantra meditation alone. Taken together, these findings suggest there may be cumulative effects of practices and chanting with intentionality may be associated with a range of psychological benefits. 

### 9.2. Intentionality and Altered States

The finding that higher self-report scores on mystical and flow states were associated with higher ratings of intentionality is consistent with research which showed mystical experiences are associated with religious beliefs [19]. However, the finding here implies that possessing specific religious beliefs may be less important than having the broader concept of intentionality during chanting. Although intentionality can be religious, it is not always. Devotion could be toward a deity in a religious sense or attached to an attitude (feelings of love or oneness) and intentions can have religious focuses or general desires (such as the desire for peace or harmony). Therefore, these findings give a broader view of underlying themes rather than specific religious beliefs. 

Findings related to the devotional aspect of intentionality align with previous evidence that spiritually meaningful phrases (during meditation) were more impactful than secular phrases at reducing anxiety, depression, and increasing spiritual well-being and pain tolerance [16,33]. Wachholtz & Pargament [33] examined whether spiritual meditation was associated with more benefits than other forms of meditation and relaxation. Participants were taught either a spiritual meditation (using phrases such as “God is peace” or “Mother Earth is love”), secular meditation (using phrases such as “I am joyful” or “I am content”) or a relaxation technique and asked to practice for 20 min per day for 2 weeks. After 2 weeks, the spiritual meditation group showed greater spiritual health (assessed with the Spiritual Wellbeing Scale) and reported more spiritual experiences than the other two groups. Specifically, the spiritual group endorsed more mystical experiences than the other two groups. Notably, in this study, the phrases used by the spiritual group indicate connecting with something other than the self (e.g., “Mother Earth is love”), whereas the phrases in the secular group are all directed towards the self (e.g., “I am content”). Therefore, this devotional aspect toward something other than the self could have been partially responsible for the enhanced mystical experiences where people often feel a sense of unity with something other than the self. These results are also consistent with preliminary research on differences in brain activity between prayers that are mechanically recited versus prayers that focus on surrendering to God [35]. Those findings suggest there are neurological differences when someone is in a state of devotion compared with when they are mechanically reciting sounds or prayer. Taken together, the findings suggest that devotional practices are associated with distinct psychological effects that may include altered states of consciousness.

### 9.3. Intentionality and Cognitive Effects

Higher self-report mindfulness and lower mind wandering scores were associated with stronger intentionality. There are several potential explanations for this outcome. Firstly, increased attentional processes may lead to greater concentration and participation in the practice. This might inhibit mind wandering, other ruminative thinking or narratives involving the self. Secondly, some chanting practices emphasize the use of mindfulness as a practice as well as an outcome [2,10,49]. That is, during the practice of chanting one does not only focus on the vocalization or mental repetition of a phrase but also maintains an objective observation of thoughts while chanting. This makes chanting a type of mindfulness training and thus would enhance mindful attention. 

Intentionality may promote self-regulation and an attitude of acceptance, thought to be fundamental aspects of mindfulness. Bishop et al. [70] proposed a two-component model of mindfulness. The first component, self-regulation is related to the ability to (a) sustain focus in the present moment, (b) switch attention from mind wandering back to the point of focus and (c) inhibit elaborate thinking (i.e., not ‘losing’ oneself in wandering thoughts). The second component of the model is an orientation or attitude of acceptance and curiosity in the present moment. Chanting might promote the two model components: self-regulation and acceptance, in a few ways. Firstly, during chanting, self-regulation may be enhanced as (a) the individual focuses on the sound which can assist with sustaining attention, (b) the sound may act as the point of focus to return to when mind wandering occurs and (c) less elaborate thinking may occur while cognitive processes are used for actively vocalizing or mentally repeating the sound. 

Intentionality while chanting may be associated with the second component of the mindfulness model, acceptance, through devotion or specific goals as many traditions promote acceptance and surrendering to events that unfold [2,7]. Thus, individuals who are devotional in their practice may be more accepting of circumstances due to their faith in a higher power. This is demonstrated by research that has found surrender to God to be associated with reduced stress and a ‘letting go’ of one’s personal wishes [71,72]. Thus, intentionality while chanting could promote mindfulness and decrease mind wandering through a range of cognitive and physiological processes. 

### 9.4. Engagement

As predicted, higher chanting engagement was associated with higher scores on altered states and cognitive benefits. Similarly, to intentionality, higher engagement could be associated with attention. However, higher engagement might allow for effortless attention, when an individual focuses completely on an activity, but the experience is one of focus without perceived effort [73,74]. Higher engagement could also mean higher dose of chanting which could enhance psychological effects simply by the increased quantity of chanting. Previous research by Bormann et al. [75] has found more chanting led to greater improvements in anxiety, religious well-being, and spiritual well-being. 

### 9.5. Engagement and Altered States

In terms of engagement, altered states were associated with longer practice duration and more regular practice but were not associated with increased experience. This finding suggests expertise may not be necessary to enter mystical or flow states, but dosage may be important over shorter periods of time. However, it is also important to note that this study focused on experienced participants who had been chanting from 1 to 725 months (*M* = 129, *SD* = 134.6) with the average being at least 10 years of chanting experience so it is unclear whether altered states might occur for complete novices to the practice. 

Longer practice duration could be associated with experiences of flow as these states likely require considerable levels of familiarity and prior experience before profound shifts toward effortless concentration can occur. Flow has been described as a state in which people feel so involved in an activity that anything other than the activity does not seem to matter [44]. Therefore, higher focus and engagement in the practice of chanting may have contributed to flow states allowing individuals to merge with the activity of chanting and forget about other worldly concerns. In addition, longer practice duration may also allow for brain changes to occur during altered states such as deactivation of the default mode network (DMN) related to flow [76] and mystical states [54]. This deactivation in the DMN has been found to be related to a reduced sense of distinction between self and other which could be related to feelings of unity experienced during these states [54]. 

A more regular practice may have also increased opportunities for effortless attention to occur. Although flow states are often associated with challenging tasks, the practice should become effortless [77]. In states of flow, challenges should be either matching or marginally higher than the individual’s skillset [73]. When individuals enter flow states, the attentional focus narrows to the stimulus (the sound in the case of chanting) and an experience of merging with the activity occurs where there is no division of attention and any events that are not related to the activity cease to be noticed [73]. Therefore, a regular practice may be necessary to develop skills in the activity of chanting to experience states of flow.

Research has also found focused attention meditation (similar to mantra meditation but focusing on an object or the breath, instead of a sound) is related to activation of brain regions involved in attention, such as the visual cortex, intraparietal sulcus and superior frontal sulcus [64]. Using functional MRI, Brefczynski-Lewis et al. [64] found experienced meditators showed less activation in these attentional regions of the brain than inexperienced meditators while practicing focused attention meditation. This suggests that a regular practice may lead to needing less attentional resources while focusing on an object, thus enhancing chances of effortless attention. Taken together, these findings suggest that longer and more regular chanting practices may eventually lead to effortless attention and may in turn be associated with altered state of consciousness.

### 9.6. Engagement and Cognitive Benefits

More experience and a more regular practice were associated with increased mindfulness and reduced mind wandering. It is likely that participants who chant regularly and are more experienced train attentional skills while chanting (such as objective awareness of thoughts and disengaging in mind wandering). Skills such as the ones described in the two-component model of mindfulness might need to be developed and thus require more experience and a regular practice to develop [70]. Taken together, these findings suggest mindfulness and mind wandering may be more responsive to a more regular practice rather than a longer practice.

### 9.7. Does Chanting Impact upon Quality of Life?

Chanting includes a variety of mechanisms that may contribute to enhanced quality of life. Overall perceived quality of life such as physical, mental and emotional health as well as quality of relationships and feelings of connection may be impacted by physiological, psychological and social aspects of chanting as well as spiritual belief systems. Physiological benefits of chanting may occur from the impact vocalization has on the nervous system, slowing breathing and enhancing feelings of relaxation [29,63]. Enhanced cognitive skills could be related to attentional aspects of focusing on sounds and thereby increase mindful attention and decrease ruminative or negative though patterns. Social aspects of chanting may enhance group cohesion from synchronization and shared belief systems. Lastly, the intensity of a range of chanting inputs may enhance mechanisms of chanting such as greater attention, engagement, and dose effects, thereby enhancing quality of life through altered states and cognitive effects. 

This study aimed to determine if attributes of chanting such as intentionality and engagement, as well as psychological outcomes of chanting collectively related to quality of life. A model was examined to explore intentionality (sound, devotion, and intention) and engagement (experience, practice duration, regularity), and how these inputs interacted with psychological outcomes (altered states and cognitive benefits) and related to quality of life. Quality of life included aspects such as improved emotional, spiritual, physical, and mental health as well as improved relationships, life satisfaction and purpose. Results showed higher intentionality and engagement impacted quality of life through altered states and cognitive benefits with flow being the biggest predictor of quality of life. These results will now be explored in depth.

### 9.8. Quality of Life and Altered States

Increased self-report of altered states was associated with quality of life. This finding is consistent with previous evidence that mystical states are related to meaning, purpose and persisting positive changes in life satisfaction [41]. Altered states provide experiences of unity and interconnectedness that could also promote quality of life [38,53,55,78,79]. Furthermore, such experiences can transform the way individuals see the world and often results in lasting positive behaviors [41,79]. Increased self-report of mystical experiences was related to quality of life primarily from intentionality (sound and devotion) and engagement (practice duration). This is aligned with previous research that found spiritual beliefs to be associated with mystical experience and confirms research that has found mystical states to be associated with purpose and overall life satisfaction [19,41]. 

Flow was associated with quality of life primarily from intention and practice duration. Flow has been found to be associated with happiness and wellbeing [47] as well as high conscientiousness and low neuroticism [77]. People have reported they are their most productive, happy and creative in states of flow [44]. Moreover, it has been argued that flow states are different to other altered states, in that, they optimize ordinary states of consciousness rather than trying to escape them [73]. Although flow states share features with other altered states (such as the merging of action and awareness, loss of time, and reduced self-referential thought), flow states can be experienced in many diverse, everyday mundane tasks (such as dancing, surgery, sport, music listening). Therefore, flow states may enhance enjoyment of everyday mundane tasks which may make them an altered state that can be integrated into one’s life more easily than other altered states that may require particular circumstances or environments. Results indicating that flow was the most impactful variable on quality of life sheds light on the value of creating certain conditions during chanting (e.g., setting intentions and longer practice duration) to enhance flow states. 

### 9.9. Quality of Life and Cognitive Benefits

Higher self-reported mindfulness related to quality of life from intentionality (intention) and engagement (experience, regularity). Whereas lower self-report mind wandering was associated with quality of life through engagement (experience and regularity). Mindfulness allows for the observation of oneself as an outside observer, often referred to as decentering or the observing self [80,81] This could enhance better perceived quality of life through disidentification with emotions and thoughts. In addition, mindfulness has been found to be associated with improved self-regulation [80], well-being [74], altruism and prosocial behavior [82,83] which in turn could promote quality of life. Whereas mind wandering has been found to be associated with states of depression which could diminish quality of life [51]. Therefore, higher mindfulness and lower mind wandering should be associated with better perceived quality of life. 

Increased mindfulness and decreased mind wandering were associated with quality of life mainly from experience and regularity. This could occur as increased mindfulness and decreased mind wandering may be acquired skills that are enhanced through focused attention practices such as chanting. Lastly, individuals who practice more regularly might obtain greater benefits than someone who practices less regularly simply due to the increased dose of the practice as found in previous research [14].

### 9.10. Chanting Style: Call and Response vs. Repetitive Prayer

Another aim of the current study was to investigate whether different chanting styles give rise to distinct psychological effects. Distinct chanting styles often emphasize different mechanisms (See Perry et al. [19] for a model of chanting mechanisms). For example, call and response style chanting is often practiced aloud, in groups, includes musical accompaniment and synchronous body movements [6]. Whereas repetitive prayer can be practiced vocally or silently and is typically done alone. In general, some outcomes were more strongly associated with specific practices compared to others. For example, an exploratory analysis suggested that call and response chanting was associated with comparatively high mystical experience scores, and repetitive prayer was associated with comparatively lower mind wandering scores. 

Participants who engaged in call and response chanting reported higher mystical experience scores than those who engaged in repetitive prayer. Whereas repetitive prayer has no musical content, call and response chanting is usually accompanied by instruments. Brancatisano et al. [84] proposed the Therapeutic Music Capacities Model (TMCM) linking the properties of music to cognitive and psychosocial benefits. The model identifies seven fundamental capacities of music that lead to mechanisms which in turn promote psychosocial benefits. The authors state that music is engaging, emotional, physical, synchronous, personal, social, and persuasive. These capacities of music may have been partially responsible for the differences found in mystical states scores of individuals engaged in call and response chanting compared to repetitive prayer. 

Other structural aspects of music may have been related to mystical states such as expectancy violations. That is, individuals have expectations of music and if such expectations are violated, emotional responses may occur [85]. Whereas in call and response chanting, the music starts slow, builds to a faster tempo and then slows again (often with a sudden drop in tempo), repetitive prayer is a much more controlled practice where the person chanting is maintaining what is usually a steady, monotonous tone and rhythm. It has been argued that violations in expectancy such as the ones in call and response chanting have been found to elicit emotional responses in music [85,86]. Taken together, call and response chanting may have been associated with mystical states more than repetitive prayer as it involves social, synchronous musical activity and may have included expectancy violations. 

Further, shared repetitive movements and vocalization during call and response chanting may allow for interpersonal synchrony and perception-action coupling, coordinating movement and visual cues [87]. Coordinated movements and vocalisations may promote brain entrainment, whereby neurons oscillate coherently across individuals’ brains, as found in previous research during coordinated speech [88]. Additionally, interpersonal synchrony and perception-action coupling has been found to be related to neurohormones that underlie feelings of social connection, such as oxytocin [89,90] In this way, interpersonal synchrony can increase positive mood and promote reciprocal prediction of the goals of others, enabling the adaptive achievement of shared goals [87]. These shared goals may promote a sense of merging with others or external stimuli, which is a common experience during mystical states [54]. 

Repetitive prayer differs from call and response in that it does not include the same social aspects, movements, or interpersonal synchrony. The absence of these elements may mean that repetitive prayer is not associated with increased oxytocin levels. Individuals practicing repetitive prayer may have shared goals with their community, but they do not synchronize these intentions with others. In contrast, call and response chanting involves coordinated intentionality, and hence may produce a stronger sense of connectedness with others (and higher oxytocin levels). 

Further, repetitive prayer may require higher levels of concentration, and more discipline (compared with call and response), which may support cognitive focus. Interestingly, participants who practiced repetitive prayer (either silently or vocally) reported lower mind wandering scores than participants who practiced call and response chanting. This is aligned with research that has found breath counting to be associated with less mind wandering and more meta-awareness [91]. 

More generally, different structural aspects of music may affect mind wandering. Research suggests that music with low complexity is associated with decreased activation in the DMN [92], an area of the brain found to be associated with mind wandering [93]. Dvorak & Hernandez-Ruiz [94] compared four different music stimuli ranging in complexity and reported that the least complex music was the most effective accompaniment to mindfulness meditation. However, when absorption was considered [95], the effect disappeared. Thus, although mind wandering was found to decrease in repetitive prayer, it is possible that absorption or other personality traits are responsible for this effect. Future research should assess the relevance of such individual difference variables in the effects of chanting. Regardless, the current findings suggest that simpler monotonous repetition of sounds may be preferable to the more complex style of call and response chanting for cognitive benefits.

It is important to note, that while many individuals practice more than one chanting style, we asked participants to report the style they engaged with most often. Individuals who show preferences for call and response chanting may have different personality characteristics than those who are inclined to practice repetitive prayer. For example, extroverted personality types may embrace social forms of chanting more than introverted individuals as they are known to seek out social activities [96]. Further, trait empathy and agreeableness are associated with the ability to perceive and entrain to external rhythms, making these styles more attractive to those with high empathy and agreeableness [97]. To our knowledge, research has yet to determine whether people with certain personalities gravitate to specific chanting styles. However, research has found that absorption, altruism, and religiosity are higher among those who experience mystical states while chanting [19]. Absorption and empathy are also associated with spiritual experiences [98]. Taken together, these results suggest that individuals with certain trait characteristics may be drawn to specific chanting practices. Furthermore, some traits may be associated with enhanced experiences of chanting. Therefore, from the correlational nature of this data, we are unable to determine whether the observed differences between call and response chanting and repetitive prayer arose because of differences in the nature of the chanting practice, personality factors, or other contextual variables.

## 10. Conclusions

This study was a widely distributed survey with 456 participants from 32 countries and varied chanting traditions. As with all surveys, the results are susceptible to potential biases. The sample was self-selected, so may not be representative of the vast population of individuals who engage in chanting practices. Moreover, the use of an online survey may have created bias limiting our sample to certain groups with access and/or competence in using computers. In addition, the survey was not translated into other languages and thus only available to English speakers. This limitation means that the participant sample is not representative of the vast and diverse population of individuals chanting around the world. Another limitation of this study was the inability to compare our sample to a similar group that did not practice chanting. Future research could compare participants who chant with a matched group of participants who do not chant. It may also be worth examining whether intentionality is related to intrinsic or extrinsic motivated goals, given that flow is associated with autotelic personality types whereby the person has a desire for performing activities for intrinsic reasons. For people with an autotelic personality, the practice has an intrinsic value and purpose in itself [77]. 

To conclude, this investigation focused on the relationship between chanting attributes (such as the nature, quality, and intensity of chanting practice) and psychological outcomes (such as altered states, cognitive benefits, and quality of life), to explore underlying determinants of chanting experiences common among diverse traditions and faiths. This study showed higher intentionality and chanting engagement were associated with altered states and cognitive benefits that in turn were related to overall quality of life. In addition, the current study shows distinct benefits are associated with various styles of chanting. While some practices are associated with alterations in consciousness, other styles are associated with higher cognitive benefits. If styles of chanting and the impact of input variables are known to mediate other psychological benefits, it may be possible to predict which type of chanting, intentionality and engagement is useful for specific psychological outcomes.

## Figures and Tables

**Figure 1 brainsci-12-01456-f001:**
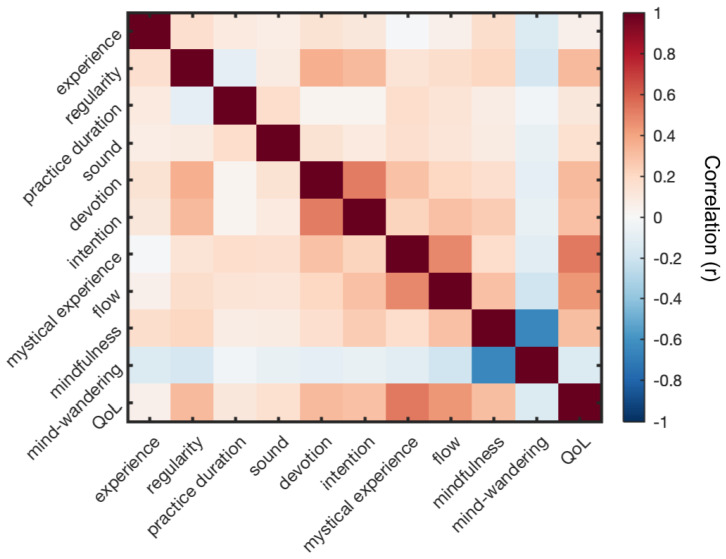
Correlation Matrix of Attributes of Chanting and Psychological Outcomes.

**Figure 2 brainsci-12-01456-f002:**
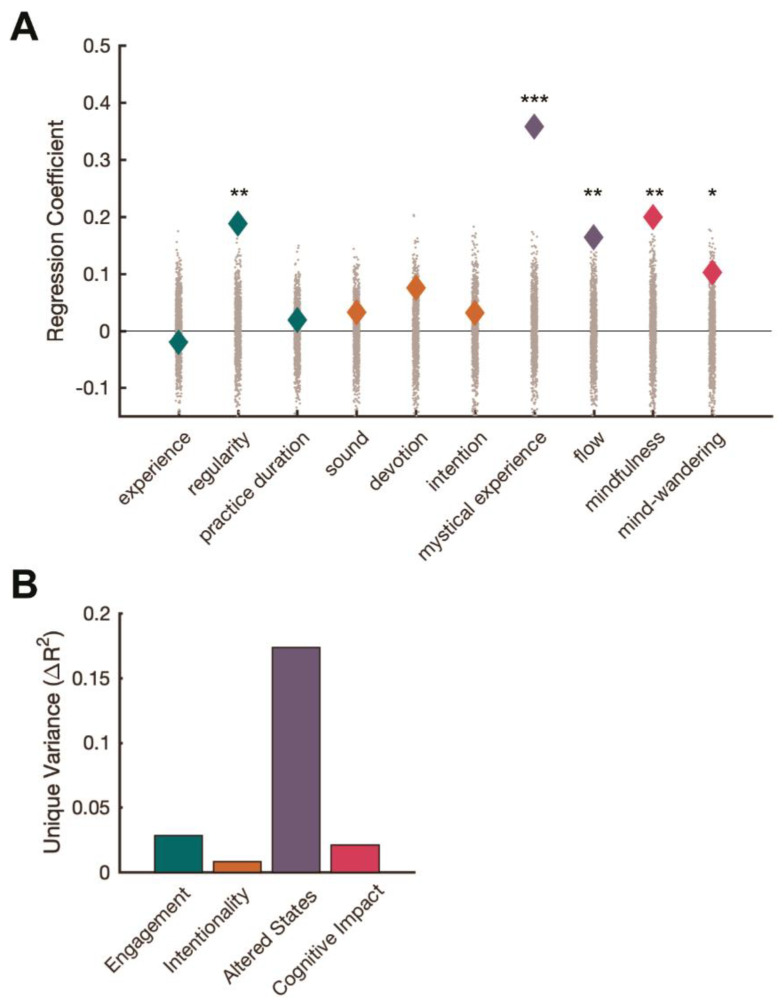
Multiple Regression Model. (**A**) showing ten predictors of Quality of Life and (**B**) showing the unique variance in Quality of Life explained by four families of variables: Engagement, Intentionality, Altered States and Cognitive Impact. Note: * = *p* < 0.05; ** = *p* < 0.01; ***= *p* < 0.001).

**Figure 3 brainsci-12-01456-f003:**
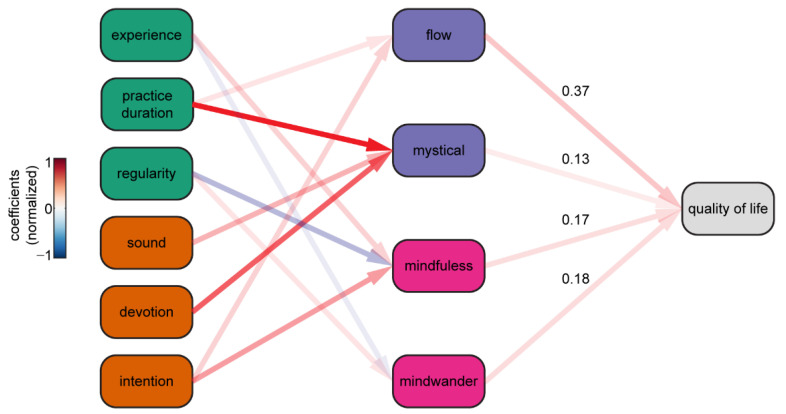
Path diagram showing that Engagement and Intentionality are indirectly associated with Quality of Life through their association with Altered States (Flow and Mystical Experience) and Cognitive Impact (Mindfulness and Mind Wandering).

**Table 1 brainsci-12-01456-t001:** Mean and Standard Deviation Scores by Chanting Style.

	Cognitive Impact		Altered States		
	Mindfulness	Mind Wandering	Mystical Experience	Flow	Quality of Life
Call and Response	61.16 (11.47)	16.21 (4.86)	108.90 (24.01)	35.00 (4.07)	72.26 (8.06)
Repetitive Prayer	63.01 (12.03)	15.09 (4.49)	103.84 (27.15)	34.69 (4.58)	73.35 (8.41)

**Table 2 brainsci-12-01456-t002:** Results of *T*-Tests Showing Differences Between Call and Response vs. Repetitive Prayer.

	Statistic	*df*	*p*	Lower	Upper	Cohens D
Mindfulness	0.602	410	0.112	−4.282	0.451	0.16
Mind Wandering	1.261	410	**0** **.020**	0.176	2.030	0.24
Mystical Exp *	1.452	347.821	**0** **.041**	0.227	10.353	0.19
Flow	2.177	410	0.492	−0.572	1.187	0.07
Quality of Life	0.682	410	0.243	−2.665	0.678	0.13

Note. Significant effects are highlighted in bold. * = Levene’s test indicated unequal variances so adjusted degrees of freedom were used.

## Data Availability

The data presented in this study are openly available in [OSF] at doi:10.17605/OSF.IO/3JFB6 (accessed on 3 October 2022).

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
