# Peer review of "How Chanting Relates to Cognitive Function, Altered States and Quality of Life"

_brainsci, 2022, doi:10.3390/brainsci12111456_

Round 1

Reviewer 1 Report

The authors of the manuscript study a very understudied yet important area in psychology, that is the connection between mysticism/spirituality and cognition. They do so by conducting a cross-sectional survey focused on chanting whereby they administered a battery of measures to 456 English-speaking participants across 32 countries. The authors applied a series of linear regressions and path analyses to show that (a) intentionality and chanting engagement is positively associated with altered states and cognitive benefits; (b) call-and-response practice is positively associated with mystical experiences; and (c) repetitive prater was associated with less mind-wandering. Overall, the manuscript represents an interesting first step in understanding the connection between chanting and psychology. However, the authors can improve the manuscript by taking the discussion a step or two further than they do by talking about the brain basis of the associations and the role of individual differences. Furthermore, the authors should walk back some claims that indicate causation when their study was correlational. 

First, some of the authors’ claims indicate causation (e.g. “contributing” in the title and “predicting” in the manuscript). This may be problematic for several reasons. The author’s data is entirely correlational; therefore, causal claims cannot be made. Furthermore, there are several important factors that may underlie their main results which make causal inferences even more problematic. For example, several of their fundings may be accounted for by the personality trait openness to experience which is known to be associated with spirituality (and aesthetic preferences and experiences). Trait conscientiousness, trait extraversion, and trait agreeableness may be other personality traits that may also come into play in distinguishing the effect of call-and-response vs repetitive prayer. Personality traits and other individual differences may provide important moderating and mediating effects that help to explain the current effects found. Though performing additional data collection and analysis should not be required for publication, the role of individual differences should be mentioned in the discussion, and any claims that infer causality should be changed. 

 Second, I was surprised that the authors did not provide more theoretical background and discussion regarding the brain basis of meditation and music. Particularly in the discussion, I would have liked to have seen more information and possible hypotheses about brain regions and networks that might be involved in chanting, and how this might differ for individual prayer vs call-and-response in groups. For example, the authors cited some of Richard Davidson’s work on attention, but not the interesting brain research he has done on meditation. 

Third, I think the authors can provide greater attention to the underlying mechanisms in play for individual prayer vs call-and-response methods of chanting. Particularly the interpersonal social and group processes that are present for call-and-response methods vs the more intrapersonal mechanisms at play during individual chanting and prayer. 

Fourth, in the introduction, the authors can expand their scope when discussing chanting to include chanting practices from other religions and practices and rituals outside of Hinduism and Buddhism, this includes Gregorian chants, Sufi chants, and chanting from the Jewish tradition. 

Fifth, the authors should specify in the abstract that they only studied English speakers. This should also be included as a limitation. It is a bit misleading to say that the study was “across 32 countries” when (a) they were all English speakers and (b) there was likely a very small representation of these 32 countries given the small sample size of 456. 

Sixth, on a similar note, though the authors provide the percentage of participants from the most largely represented countries, given that the authors emphasize “32 countries", I think readers would want to see in supplemental material the distribution of participants for each of the 32 countries. 

Seventh, it would be beneficial for the authors to include more demographic information for the participants, especially, for ethnicity and religious affiliation. 

Eighth, I’d be curious how the results might change if the authors conducted a PCA on some of the variables (e.g. the attributes in Figure 2A). Though this isn’t necessary for publication, I’d be curious how such variables group together and if they add more clarity and transparency to the results. 

Nineth, there is a big drop off from the 707 people who the authors say clicked the survey link and the 456 participants who completed the entire battery. This raises questions about whether the survey was too long. And if so, are there biases based on the type of personality who completed the survey (e.g., higher in Conscientiousness). Adding a statistic here on the average number of questions completed by the entire 707 people and the drop off rate might be helpful. Or perhaps the majority of the 707 people simply clicked the hyperlink but didn’t begin the survey, which would indicate less confounds. 

Tenth, the authors should do another grammatical check (e.g., p. 7 “5-poin Likert”). 

Reviewer 2 Report

The article entitled “Mindfulness, Mysticism and Mantra: How Chanting Contributes to Cognitive Benefits, Altered States and Quality of Life” describes a study that examines the cognitive and emotional features of experiential reports of chanting, including flow states, mystical experiences, mindfulness, and mind wandering. It also attempts to link such reports to quality-of-life judgments. All data come from self-reports of a reasonably large sample of participants who regularly practice chanting. The study is interesting and the article is well-written. I believe a few minor changes to the current narrative may make the paper suitable for a broad readership.

In the introduction, the authors state that their survey is intended to examine the relationship between attributes of the practice of chanting and particular psychological experiences as well as quality-of-life judgments. They also aim at understanding whether differences in chanting styles are related to differences in psychological experiences. Because the design and methodology adopted by the authors and most of the analyses they carry out define the study as correlational research, the authors would be well advised to re-write the title and the abstract to ensure that both describe relationships.  For instance, the findings of their study test hypotheses concerning relationships between or among variables. Implying cause-effect relationships may be a bit of a stretch. If the authors had formulated the hypotheses differently and assessed brain states during chanting, then cause-effect relationships could be explicitly mentioned.

At the start of the result section, the authors may want to add a section in which for each hypothesis they briefly describe the analyses performed. A summary of the analysis may help the reader to better understand the results of the study.

Figures can be informative as well as appealing, but they often lack all information that researchers need to evaluate the quality of a study. Thus, Figures 2A and 2B may need to be changed into a traditional table for reporting linear regression results, including b = unstandardized regression coefficient, SE = standard error of the unstandardized regression coefficients, Beta = standardized regression coefficients, structure coefficients, and squared structure coefficients. Also, report information about the absence of multi-collinearity (tolerance values and mean VIF; see Field, 2013). A similar suggestion to report more information applies to the path analysis part of the result section, albeit the figure here is more helpful.

 According to the authors, in Figure 3, colored arrows indicate significant effects while nonsignificant effects are marked by grey arrows. The lines can be made slightly wider to facilitate readability.  Rather than ‘significance effects’, the authors may want to mention ‘significant relationships'.
